# Human-Induced Pluripotent Stem Cell (iPSC)-Derived GABAergic Neuron Differentiation in Bipolar Disorder

**DOI:** 10.3390/cells13141194

**Published:** 2024-07-15

**Authors:** Daniel J. Schill, Durga Attili, Cynthia J. DeLong, Melvin G. McInnis, Craig N. Johnson, Geoffrey G. Murphy, K. Sue O’Shea

**Affiliations:** 1Department of Cell and Developmental Biology, The University of Michigan, Ann Arbor, MI 48109, USA; durgaa@med.umich.edu (D.A.); cjweaver@med.umich.edu (C.J.D.); johnscrn@med.umich.edu (C.N.J.); oshea@med.umich.edu (K.S.O.); 2Department of Psychiatry, The University of Michigan, Ann Arbor, MI 48109, USA; mmcinnis@med.umich.edu; 3Department of Molecular and Integrative Physiology, The University of Michigan, Ann Arbor, MI 48109, USA; murphyg@med.umich.edu

**Keywords:** GABA switch, gamma-aminobutyric acid, KCC2, NKCC1, neuron, patient, skin biopsy, *SLC12A2*, *SLC12A5*, somatostatin, stem cell

## Abstract

Bipolar disorder (BP) is a recurring psychiatric condition characterized by alternating episodes of low energy (depressions) followed by manias (high energy). Cortical network activity produced by GABAergic interneurons may be critical in maintaining the balance in excitatory/inhibitory activity in the brain during development. Initially, GABAergic signaling is excitatory; with maturation, these cells undergo a functional switch that converts GABA_A_ channels from depolarizing (excitatory) to hyperpolarizing (inhibitory), which is controlled by the intracellular concentration of two chloride transporters. The earliest, NKCC1, promotes chloride entry into the cell and depolarization, while the second (KCC2) stimulates movement of chloride from the neuron, hyperpolarizing it. Perturbations in the timing or expression of NKCC1/KCC2 may affect essential morphogenetic events including cell proliferation, migration, synaptogenesis and plasticity, and thereby the structure and function of the cortex. We derived induced pluripotent stem cells (iPSC) from BP patients and undiagnosed control (C) individuals, then modified a differentiation protocol to form GABAergic interneurons, harvesting cells at sequential stages of differentiation. qRT-PCR and RNA sequencing indicated that after six weeks of differentiation, controls transiently expressed high levels of NKCC1. Using multi-electrode array (MEA) analysis, we observed that BP neurons exhibit increased firing, network bursting and decreased synchrony compared to C. Understanding GABA signaling in differentiation may identify novel approaches and new targets for treatment of neuropsychiatric disorders such as BP.

## 1. Introduction

Bipolar disorder is one of the most severe yet poorly understood neuropsychiatric conditions. Typically diagnosed in late adolescence to early adulthood, there is a clear developmental contribution to BP, but the cellular underpinnings are poorly understood, largely due to the lack of viable, relevant cell types for study. With the recent development of methods to derive pluripotent stem cells from adult somatic tissues, it is now possible to carefully examine the sequential steps in the differentiation of glial and neuronal cells from human pluripotent stem cells to determine when and why development may go awry. Human iPSC can also form theoretically unlimited numbers of patient-matched cells for transplantation, cells can be genetically modified using, e.g., CRISPR, they can be labeled and grown in vitro to examine their responsiveness to medicines, and multiple cell types can be generated using organoids and assembloids [1,2] to increase tissue complexity. Unpatterned cells can be employed to test the activation (and inhibition) of signaling pathways, timing of gene expression and differentiation contributing to birth defects of the CNS.

A balance between inhibitory and excitatory a ctivity in the brain is critical in setting up and maintaining cortical circuitry during development, and errors in this process have been suggested to play a role in several developmental disorders of the brain, including autism spectrum disorders, schizophrenia, BP and epilepsy [3]. Considerable research has focused recently on neuronal inhibition especially in the case of, e.g., epilepsy. In fact, research has identified perturbations in GABA neurotransmission in BP, and drugs such as valproate and lithium that improve BP symptoms typically increase GABA release [4,5,6].

GABAergic interneurons form in the ventral telencephalon, then undergo long-distance tangential migration to the cortex where they associate with glial processes and migrate radially toward the cortical plate [7]. The polarity of GABA interneurons is initially determined by the relative expression levels of the chloride transporters NKCC1 (*SLC12A2*) and KCC2 (*SLC12A5*) [8]. Early in development, NKCC1 predominates, increasing intracellular chloride levels and promoting depolarizing, excitatory currents in interneurons. Later in development, as neurons increase their production of KCC2, chloride moves into the cell, hyperpolarizing it, and GABA attains its adult inhibitory function. Alterations in the timing of this “switch” from depolarization to hyperpolarization are thought to occur midway–late in human gestation; any alteration in the timing or level of expression of these transporters can disrupt the E/I balance and produce changes in neuronal migration, proliferation, plasticity and neuronal architecture [7] that have long-lasting effects on cortical organization, which may only be identified decades later. Behavioral stresses [9,10] can alter the expression of these transporters, and both premature and delayed expressions of NKCC1 [11] have been associated with a broad spectrum of neurodevelopmental disorders including autism spectrum disorder, ASD [12]; fragile X syndrome [12]; Rett syndrome [13,14]; Down syndrome [15]; BP [16]; and schizophrenia [17].

In the current investigation, we modified a neuronal differentiation protocol to produce widespread differentiation of SST+ forebrain neurons from human iPSC. We have characterized the ability of pluripotent stem cells derived from undiagnosed control individuals (C) and from patients with BP who carry a single nucleotide polymorphism (SNP), *rs1006737*, in the L-type calcium channel gene *CACNA1C*, which regulates calcium influx and has been associated with SZ [18,19], schizophrenia [20,21] and major depression [22], to mature and function. We generated eight independent lines of BP and C iPSC and used CRISPR to correct the SNP in three of the BP lines that carry *rs1006737* to develop isogenic iPSC lines.

## 2. Materials and Methods

With Institutional Review Board approval (HUM00043228), skin biopsies (3 mm) were taken from four patients diagnosed with bipolar I disorder who carry the *rs1006737* risk allele (AA) in *CACNA1C* and from four control individuals who do not carry this allele (GG), two female and two male, with no psychiatric diagnosis (Table 1). The patients are participants in a longitudinal study of bipolar disorder who were previously diagnosed with BP1 at the University of Michigan Department of Psychiatry; the controls had no psychiatric diagnosis. Sterile biopsies of patient skin were anonymized, disaggregated, plated in sterile Petri dishes and expanded for 1–3 passages prior to testing for sterility, mycoplasma and SNP genotyping. Resulting fibroblasts were expanded and reprogrammed to pluripotency by transfecting them with episomal expression vectors containing the following reprogramming factors: *SOX2*, *OCT4*, *KLF4*, *L-MYC* and *LIN28* (Epi5, Thermo Fisher Scientific, Waltham, MA, USA) with human pluripotent stem cell oversight approval (HPSCRO approval #104800). Clonal lines were selected, re-anonymized and tested for normal karyotype, expression of pluripotency markers, lineage differentiation capacity (as embryoid bodies), absence of mycoplasma and absence of the reprogramming cassette. Loss of episomal vector DNA was confirmed by quantitative PCR detection of the episomal vectors, as described in [23]. The *rs1006737* risk allele in three BP iPSC were targeted for correction via CRISPR-Cas9-mediated editing and repair at WiCell (Madison, WI, USA). Clonal lines were established and CRISPR correction was confirmed via Sanger sequencing (Figure 1A). Additionally, karyotype (Figure 1B) and mycoplasma analysis were performed following editing and cell line expansion. iPSC lines were cultured on 0.08 mg/mL Matrigel (Corning, Corning, NY, USA, #354277) in TeSR-E8 media (Stem Cell Technologies Vancouver, BC, Canada, #05990).

### 2.1. Neuronal Differentiation

iPSC lines were differentiated into GABAergic neurons, following [24] with modifications (Figure 2). Briefly, iPSC lines were cultured as embryoid bodies (EBs) with dual SMAD inhibition in DMEM/F12 containing 20% KOSR, 100 µm BME, 1X glutamax, 1X non-essential amino acids (NEAA), 2 µM A-83 (Cayman Chemical, Ann Arbor, MI, USA, #9001799) and 2 µM dorsomorphin (Cayman Chemical, #11967). After 5 days, the medium was changed to neural induction medium (NIM): DMEM/F12 containing 1X N2 (Thermo Fisher Scientific, #17502-048), 1X glutamax, 1X NEAA and 300 nM of the Sonic Hedgehog Agonist (SAG) (Cayman Chemical, #11914), and grown for an additional 2 days as EBs. Next, they were plated on 0.04 mg/mL Matrigel for 14 days in NIM to form rosettes. They were fed every other day for the first 7 days and every day for the next 7 days. Following rosette formation, neurospheres were picked and grown in suspension in NIM for 6 days. Neurospheres were dissociated using a gentleMACS dissociator (Miltenyi Biotec, Gaithersburg, MD, USA), and these neural progenitor cells (NPCs) were then plated in NIM on dishes coated in 0.1% poly(ethyleneimine) (pEI) (Millipore Sigma, St. Louis, MO, USA, P3143) and 20 µg/mL laminin (Corning, #354232). For neuronal differentiation, NPCs were cultured in neuronal differentiation medium (NDM); Brainphys neuronal media (Stem Cell Technologies, #05792), with 1X SM-1, 1X N2, 20 ng/mL BDNF (Thermo Fisher, #450-02) and 20 ng/mL GDNF (Peprotech, #450-010). At the initial medium change with NDM, 200 nM Compound E was added (Millipore Sigma #56579). One mM Bucladesine (cAMP) (Millipore Sigma, #D0260) was added at every medium change for the first two weeks of neuronal differentiation. GABAergic neurons were cultured for up to 6 weeks with half media changes twice a week; 1 µg/mL laminin was added at every other medium change for 2 weeks in culture. GABAergic neurons were characterized via immunocytochemistry (ICC), qRT-PCR, bulk RNA sequencing, scanning electron microscopy, live cell calcium imaging and multi-electrode array analysis.

### 2.2. Immunocytochemistry

Cells were fixed in 4% paraformaldehyde (Electron Microscopy Sciences, Hatfield, PA, USA, #15700) for 10 min, then washed and stored in 1X PBS (with Ca^++^/Mg^++^) (Thermo Scientific, #14040-133) at 4 °C. For immunocytochemistry, cells were permeabilized with 0.1% Triton X-100 (Millipore Sigma, T9284) in 1X PBS, blocked with 3% bovine serum albumin (BSA) (Millipore Sigma, #126575) and exposed to primary antibodies overnight at 4 °C in 1% BSA. Antibodies and concentrations employed were as follows: GAD65-67, 1:500 (Abcam, Cambridge, UK, ab49832), MAP2, 1:2500 (Abcam, ab5392), SV-2, 1:500 (Synaptic Systems, Goettingen, Germany, #119 004), NeuN, 1:100 (Millipore Sigma, MAB377), TTF1 (Nkx2.1), 1:500 (Abcam, ab76013), FoxG1, 1:500 (Abcam, ab18259) and synapsin-1, 1:250 (Cell Signaling, #5297). Secondary antibodies, added for 1 h at room temperature, all utilized at a 1:500 dilution, include the following: AF 555 donkey anti-rabbit antibodies (Thermo Fisher Scientific, A31572), AF 647 donkey anti-mouse (Thermo Fisher Scientific, A31571), AF 488 donkey anti-guinea pig (Jackson ImmunoResearch Laboratories, West Grove, PA, USA, #706-545148), AF 647 donkey anti-chicken (Jackson ImmunoResearch Laboratories, #703-605-155) and AF 488 donkey anti-rabbit (Thermo Fisher Scientific, A21206). Additionally, cells were stained with 300 nM DAPI (Thermo Fisher Scientific, #62248) prior to viewing. Images were acquired using a Leica DMI8 inverted fluorescence microscope and accompanying LAS X (Leica Version 1.4.6).

### 2.3. RNA Sequencing

After 6 weeks of differentiation, total RNA was extracted from cells using Trizol (Thermo Fisher, #15596018) followed by purification with the RNAeasy mRNA Isolation Kit (Qiagen, Redwood, CA, USA, #74104) according to the manufacturer’s instructions. RNA concentrations were calculated spectrophotometrically, and samples were stored at −80 °C. RNA aliquots containing up to 200 ng of RNA from two independent biological differentiations from 4 control, 4 BP and 3 BP-corrected samples were submitted to the BGI Group (BGI) where RNAs were labeled using the TruSeq (polyA) stranded kit and sequenced using the NovaSeq 6000 platform (Illumina, San Diego, CA, USA).

Prior to analysis, 1.27B raw sequence reads were subjected to quality control checks using FastQC (v0.11.9). Adapter sequences and low-quality bases were trimmed using Trim Galore (v0.6.6) with the recommended parameters for RNA-Seq data. Processed reads were aligned to the human reference genome (GRCh38.v40) using the STAR aligner (v2.7.10a). STAR was chosen for its ability to handle spliced alignments effectively, and we used the gene annotation file GENCODE v40 primary assembly. Aligned reads were sorted by coordinates, and the resulting SAM files were converted to BAM format using SAMtools (Version 1.6) for downstream processing.

The featureCounts utility (Rsubread package v2.16.1) was used for the gene-level quantification of aligned reads. The gene annotations provided by the GENCODE v40 reference (gencode.v40.primary_assembly.annotation.gtf URL accessed on 22 August 2022) were used to assign reads to genes. Parameters for featureCounts were set to ignore multimapping reads.

Read counts generated by featureCounts were used for differential expression analysis through the edgeR (v3.36.0) and limma (v3.50.3) Bioconductor packages in R (v4.1.2). Normalization factors to scale raw library sizes were computed using the calcNormFactors function in edgeR. The voom method from the limma package was applied to estimate the mean–variance relationship and to return log2 counts per million data suitable for linear modeling and empirical Bayes moderation.

### 2.4. qRT-PCR

cDNA was synthesized from 50 ng of RNA per sample using Superscript III reverse transcriptase (Thermo Fisher, #56575) and random primers (Invitrogen, #58875; Appendix A). Quantitative RT-PCR was performed using a Quantstudios 3 (Thermo Fisher Scientific) PCR Machine and PowerUP SYBR Green Master mix (Thermo Fisher Scientific, #A25741). RPL27 was the housekeeper for normalization. Pearson correlation was used to compare gene expression in qRT-PCR with gene expression in RNA-Seq to validate the RNA-seq data.

### 2.5. Statistical Analysish

Quantitative data including quantitative RT-PCR expression data were analyzed in SPSS using paired *t*-test or ANOVA; a *p* of ≤0.05 was considered significant. In cases with high variance, outliers were removed from the analysis if they were outside 2.2 X the interquartile distance [25,26]. We also standardized expression values by comparing expression of the targeted transcript to the value of a neuron-restricted gene, e.g., NeuN (neuronal nuclei) present in most nuclei of neurons.

### 2.6. Scanning Electron Microscopy

Cells were grown on coverslips, fixed with 2.5% glutaraldehyde in 0.1 M cacodylate buffer and transferred to the Microscopy and Image Analysis Core Facility at the University of Michigan. Cells were post-fixed in 2% osmium tetroxide, dehydrated through increasing concentrations of hexamethyldisilazane, and images were acquired on a JEM 1400 Plus TEM (JEOL).

### 2.7. Live Cell Calcium Imaging

Cells were cultured on 12 mm glass coverslips and transferred to a perfusion chamber with field stimulation (Warner Instruments, Hamden, CT, USA, RC-49MFSH). Cells were incubated for 10 min with 2 µM Fluo-5 AM (Thermo Fisher Scientific, F14222) in phenol red-free BrainPhys media (Stem Cell Technologies, #05791). Fluo-5AM-containing media were then removed, replaced with phenol red-free Brainphys alone, and cells were imaged on an Olympus 1X83 microscope with accompanying CellSens software (Olympus, Version 4.3). Cells were imaged for 1 min and 30 s, with one image acquired per 0.15 s and three image sets acquired per coverslip. At 40 s, coverslips were pulsed with whole-field stimulation at 20 Hz via an A310 Accupulser (World Precision Instruments, Sarasota, FL, USA) and S1U-102B stimulus isolation unit (Warner Instruments). Movies were generated using ImageJ and analyzed following bleach correction and background subtraction. The plugin Diff Image 1C was utilized to distinguish frames where the stimulation occurred. Up to 150 ROIs were selected via the time series analyzer v3 plugin per data set and delta F divided by F was calculated over time following correction via a normalization constant generated from fluorescent intensity of ROIs prior to stimulation. Additionally, area under the curve and peak amplitude were calculated using GraphPad Prism 8 software.

### 2.8. Multi-Electrode Array Analysis

Three hundred thousand cells per well were plated on Accuspot classic MEA 48 well plates (Axion Biosystems, Atlanta, GA, USA, #M768-KAP-48A) containing 0.1% pEI and 100 µg/mL laminin and differentiated in NDM for 6 weeks. Spontaneous activity was recorded for 5 min three days a week for the 6-week differentiation period on a Maestro MM (Axion Biosystems) and accompanying Axion Integrated Studio (AxiS) 2.4.2 software. A 200 Hz to 3 kHz filter and 5.5 × STD spike detector were utilized to collect raw and spike files. Plates were acclimated to the machine for up to 3 min prior to recording. Half media changes with NDM were performed following recordings. At the initial feed with NDM 200 nM, Compound E was added (Calbiochem #56579, Sigma:Aldrich, St. Louis, MO, USA), and 1 mM Bucladesine (cAMP) (Millipore Sigma, #D0260) was added at every feed for the first two weeks of neuronal differentiation. Following 2 weeks of differentiation, 1 µg/mL laminin was added to NDM once a week. Data were analyzed using the accompanying AxiS Metric plotting tool software (Axion Biosystems). Weighted mean firing rate, number of network bursts and area under normalized cross-correlation (synchrony) are presented, and other parameters are available upon request.

## 3. Results

### 3.1. Generation and Validation of CRISPR-Corrected Isogenic Bipolar Disorder Cell Lines

Cell lines derived from BP patients in this study harbor a single nucleotide polymorphism, *rs1006737*, in the L-type calcium channel gene *CACNA1C*, which mediates calcium influx into the cell and has been associated with BP as well as with schizophrenia. However, the impact *rs1006737* has on development and function of GABAergic neurons has not been conclusively determined and may be region- or cell-type-specific. Three bipolar patient cells isogenic lines were generated to investigate the effects of *rs1006737* on neural development. A total of 11 cell lines were utilized, four control (C), four bipolar disorder (BP) and three isogenic CRISPR-corrected bipolar disorder (BP-C) cells (Table 1). These cell lines demonstrated conversion of the rs*1006737* SNP from A to G by Sanger sequencing (Figure 1A) and normal karyotype (Figure 1B). Representative images of BP-C iPSC show expressions of the stem cell marker genes OCT4, Nanog, TRA-160 and SSEA-1 (Figure 1C), demonstrating their pluripotency.

### 3.2. GABAergic Neuron Differentiation

To identify potential differences in development and cellular function in BP vs. C neurons, we generated GABAergic neurons using the protocol outlined in Figure 2A, modified from [24]. Given recent reports that the smoothened agonist SAG activates the SHH pathway to promote both neuronal differentiation and survival [27,28], we replaced SHH with SAG. GABAergic neurons were harvested for downstream analyses including qRT-PCR (Figure 2B), immunocytochemistry (Figure 2C), scanning electron microscopy (SEM) (Figure 2D), bulk RNA sequencing (Figure 3), multi-electro array (MEA) (Figure 4) and calcium imaging (Figure 5), following 6 weeks of neuronal differentiation. qRT-PCR analysis demonstrated that after 6 weeks, control (C), bipolar (BP) and BP-corrected (BP-C) GABAergic neurons express GAD67 (Figure 2B), a glutamate decarboxylase that converts glutamate to GABA. There was an increase in GAD67 expression in BP and in CRISPR-corrected (BP-C) GABAergic neurons compared to control, but this increase was not statistically significant (Figure 2B). These cultures also contained low levels of S100β present in astrocytes and the glutamatergic marker _v_Glut1 (Figure 2B). Next, we performed immunocytochemistry to further characterize the cells (Figure 2C). The GABAergic marker GAD65/67, NK2.1 for MGE, Foxp1, a forebrain marker, and microtubule-associated protein 2 (MAP2) neuronal marker were expressed in 6-week C, BP and BP-C 6-week GABAergic neurons. Lastly, 6-week GABAergic neurons were subjected to SEM. C, BP and BP-C neurons demonstrated normal neuronal morphology with no gross differences observed except for an increased density present in BP-C cultures (Figure 2D). From these data we concluded that we can successfully generate cultures highly enriched (>85%) in GABAergic neurons, and that there are minimal differences observed in BP and BP-C as compared to C. Next, we utilized these neurons to investigate differences in cellular components and neuronal function over developmental time.

### 3.3. Bulk RNA Sequencing Analysis of GABAergic Neurons

Bulk RNA sequencing was performed on 6-week GABAergic neurons to determine if there were differences in gene expression at this time in RNA from BP neurons. Two independent differentiations of the 11 cell lines were performed and principal component analysis (PCA) carried out. PCA identified clustering of samples from the same genetic background, demonstrating consistent differentiation, with weak clustering when comparing different genetic backgrounds in either C, BP or BP-C groups (Figure 3A). One C5G sample was removed from this analysis due to poor quality control data after sequencing. Volcano plots comparing BP and C GABAergic neurons show numerous differentially expressed genes with several genes that have been shown by others to contribute to BP, another neurological disorders, to GABAergic neuron differentiation or to general neuronal development (Figure 3B). Genes that are upregulated in C compared to BP are highlighted in blue, and genes upregulated in BP compared to C are labeled in red.

Volcano plots comparing C to BP-C and BP to BP-C similarly show differentially expressed genes (DEG, Figure 3B). Lastly, a heatmap of the top 25 genes from BP vs. C, C vs. BP-C, and BP vs. BP-C identified 62 unique genes, 60 of which are displayed (Figure 3C). In this gene set, BP and BP-C demonstrate similar sets of genes up- and downregulated compared to their C counterparts (Figure 3C), suggesting that there are similarly and differentially expressed genes present in BP, BP-C and C 6-week GABAergic neurons. Among these are transcripts encoding extracellular matrix components, synapses and ion channels. Like genes differentially expressed in the heatmap, these transcripts may identify novel pathways and treatment targets for BP (Appendix A).

### 3.4. Multi-Electrode Array Analysis of GABAergic Neurons

To begin to investigate the cellular function (s) of GABAergic neurons from BP patients, we performed multi-electrode array (MEA) analysis on neurons with increasing time in culture. Spontaneous activity was recorded for 5 min three days a week for the entire 6-week differentiation period. Beginning on day 10 and continuing to day 50 of GABAergic neuronal development, BP neurons demonstrated a statistically significant increase in their mean firing rate and network bursting compared with BP-C and C neurons (Figure 4A,B). Additionally, between days 0 and 30 of GABAergic neuron differentiation, BP neurons showed a statistically significant increase in their network bursting compared with BP-C and C neurons (Figure 4A,B). Lastly, between days 0 and 40, C neurons exhibited a significant increase in synchrony compared to BP-C and C neurons (Figure 4A,B).

### 3.5. Live Cell Calcium Imaging of GABAergic Neurons

To investigate the cellular differences in GABAergic neurons in bipolar disorder compared with control neurons, we performed live cell calcium imaging on 6-week neurons. Cell lines were derived from carriers of a single nucleotide polymorphism, *rs1006737*, in the L-type calcium channel gene *CACNA1C*, which has been associated with both BP and schizophrenia. GABAergic neurons were differentiated for 6 weeks and pre-incubated with Fluo-5-containing media. During imaging, cells were subjected to 20 Hz stimulation (Figure 5A) and changes in fluorescence intensity, delta F divided by F, were calculated over time for up to 150 regions of interest (ROIs) per image. Comparing 6-week C, BP and BP-C GABAergic neurons, peak amplitude and area under the curve in response to stimulation were slightly increased in both BP and BP-C as compared to C, but this did not reach statistical significance (Figure 5B,C). From these data, we conclude that calcium handling was largely unaltered in 6-week BP GABAergic neurons.

### 3.6. Expression of Transporters during Differentiation of BP Patient GABAergic Neurons

To identify patterns in transporter gene expression over time, we first used descriptive statistics to determine when during the six-week culture period there were significant differences between groups (Appendix A). The first indication of change was the difference in KCC2 expression in BP vs. C neurons identified as early as week 1 of differentiation, persisting into week 3. The next major changes were in NKCC1 expression differences between BP and C neurons identified in week 4 (*p* ≤ 0.016) persisting to week 6 (*p* ≤ 0.032). Differences in the ratio of NKCC1 to KCC2 between BP and C were observed on week 3 (*p* ≤ 0.047) and were also present during weeks 4 (*p* ≤ 0.055) and 6 (*p* ≤ 0.065).

In our control neurons, qRT-PCR analysis (Figure 6) showed that NKCC1 expression remained relatively low for the first three weeks in culture, after which time (week 4), expression increased rapidly, remaining elevated on weeks 4 and 5 before dropping on week 6 (Figure 3). This could maintain a stronger depolarizing GABA environment in C compared with BP neurons to promote proliferation, neurite outgrowth, cell migration and synaptogenesis. BP neurons produced similar levels of NKCC1 as controls during the first 3 weeks, NKCC1 expression increased to a moderate peak on week 5 before dropping. In BP neurons, KCC2 gradually increased as activity was recorded for 5 min three days a week for the entire 6-week differentiation period. Beginning on day 10 and continuing to day 50 of GABAergic neuronal development, BP neurons demonstrated a statistically significant increase in their mean firing rate and network bursting compared to BP-C and C neurons. Additionally, between days 0 and 30 of GABAergic neuron development, BP neurons demonstrated a statistically significant increase in their network bursting compared to BP-C and C neurons (Figure 4A,B). Lastly, between days 0 and 40, C neurons exhibited a statistically significant increase in synchrony compared to BP-C and C neurons (Figure 4). This pattern would result in considerably less exposure of C neurons to depolarizing GABA, which could decrease proliferation and neurite outgrowth in BP compared with C neurons. In controls, KCC2 was expressed at low levels for the first 4 weeks followed by a moderate peak at week 6. Increasing KCC2 would be expected to inhibit NKCC1 expression, eliciting hyperpolarizing/inhibitory currents. This would switch cell proliferation, process outgrowth and cell migration to circuit consolidation and sensory input. Current research suggests that the ratio of NKCC1/KCC2 may provide a method to fine-tune the E-I balance, as well as to control morphological differentiation [29]. High NKCC1/KCC2 ratios have been associated with several neurodevelopmental conditions, including psychiatric, neurological and birth defects caused by abnormalities of cell proliferation, migration, dendrite differentiation, etc. Both our BP and control neurons were characterized by increasing ratios of NKCC1/KCC2, although the BP group had a lower peak.

## 4. Discussion

The ability to induce adult somatic (in this case skin) cells to behave as pluripotent stem cells (iPSC) offers an unsurpassed opportunity to model human disease and is providing exciting new data regarding human development at stages not otherwise available for study. In the case of neuropsychiatric disorders such as bipolar disorder, iPSC provide a source of viable cells that can be differentiated into the many cell types of the brain to study their role in BP and other neurodevelopmental disorders. Obtaining samples from individuals previously diagnosed with bipolar disorder has the significant advantage of allowing fibroblast derivation to be prioritized based on diagnosis, age, drug response, genetics and/or disease progression.

GABA is one of the first neurotransmitters active in the brain and is the most abundant and most widely distributed inhibitory neurotransmitter in the cortex [30], placing it in a pivotal position to direct neuronal behavior during development. The *solute carrier 12 (SLC12*) gene family contains two major Cl^−^ co- transporters, NA^+^-K^+^-2Cl^−^ co-transporter 1 NKCC1 (*SLC12A2*) and the K^+^-Cl^−^ co-transporter 2 KCC2 (*SLC12A5*), which control GABA currents; the timing of their expression, region- and cell-type-restricted expression determine the direction of GABA currents and thereby, the long-term organization and activity of the cortex (8). During periods of early depolarizing GABA activity when NKCC1 predominates, neurons are prompted to divide, undergo process outgrowth, cell migration and synaptogenesis. GABAergic neurons increase production of KCC2, which exports chloride from the neuron hyperpolarizing it; with time, these neurons attain their mature inhibitory function. Since several neurodevelopmental disorders appear to result from the misregulation of NKCC1, NKCC1 has itself become a therapeutic target. One inhibitor of NKCC1, the diuretic bumetanide [31,32,33], has been particularly useful in this regard, although it passes poorly through the blood–brain barrier. It would be of particular interest to examine its effects and test additional inhibitors in the early phases of differentiation of these neurons. During periods of hyperpolarizing GABA, synaptic circuits are established and reinforced, and it may be of interest in this regard that in our research, neurons differentiated from BP iPSC are characterized by reduced synaptic densities. An additional observation that may also be of relevance to BP, given the role of stress in eliciting BP episodes, is that early life stress including maternal separation [9,10] has been shown to decrease KCC2 levels in the hippocampus and delay the GABA shift.

Studies of other disease-carrying neurons have identified a “defective” ratio of NKCC1:KCC2 [12]. However, the ratio remained low throughout the development of our BP neurons. The cause of this is unknown, although the striking increase in NKCC1 production in weeks 4 and 5 may be involved. It is also possible that the addition to the medium of BDNF (which binds Trkb receptors and promotes the secretion of NKCC1), thereby accelerating the GABA shift [34], could be involved. Support for this, however, is reduced by the fact that BP cultures received similar amounts of BDNF without apparent stimulation of NKCC1.

Early in development, calcium movement in the cell plays an important role in the morphogenesis of the CNS, affecting gene expression, cell division, migration, process outgrowth and synaptogenesis. A SNP *rs1006737* in exon 3 of Ca_v_1.2, which is highly associated with bipolar disorder, schizophrenia and depression, was recently identified in GWAS. In the current investigation, we have taken advantage of the large cohort of patients (n > 1600) participating in a longitudinal study of bipolar disorder to obtain skin biopsies from individuals who carry this SNP to develop iPSC lines that can be differentiated into neurons and glial cells. In addition, we used CRISPR to develop isogenic cell lines from the same bipolar disorder patients in which the G to A polymorphism in CACNA1C has been repaired. We characterized gene expression profiles using bulk RNA-seq and qRT-PCR and functional capacity using MEA to understand the developmental and functional differences in calcium handling and GABA signaling in these neurons. Bulk RNA-seq analysis identified significant differences in the expression of classes of genes that may play a role in the differentiation of GABAergic neurons in bipolar disorder and in nervous system development. Several of these have previously been implicated in bipolar disorder or early development (Appendix A).

Dysregulation of calcium homeostasis has long been hypothesized to underly neurodevelopmental and neuropsychiatric conditions based on observations that there were increased levels of calcium in blood cells of BP patients [35]. While much remains to be learned about the role of *rs1006737* in calcium homeostasis and early CNS development, an in vitro system where differentiation can be sequentially monitored should significantly improve our understanding of the early events in CNS development and how normal development may go awry.

Limitations of the study include the small sample size typical of human pluripotent stem cell research due largely to costs and patient availability. In the current investigation, this is partially mitigated by CRISPR correction of the SNP, allowing us to develop isogenic lines, thereby decreasing variance. A concern with this approach can also be the damage associated with the chemicals used in CRISPR selection. Another limitation is that while these cultures contain an enriched population of neurons, they lack cell types preset in the developing nervous system that contribute to the full complement of brain cells. These cells should provide a source of GABA+ interneurons for transplantation, disease modeling and cell replacement therapies.

## Figures and Tables

**Figure 1 cells-13-01194-f001:**
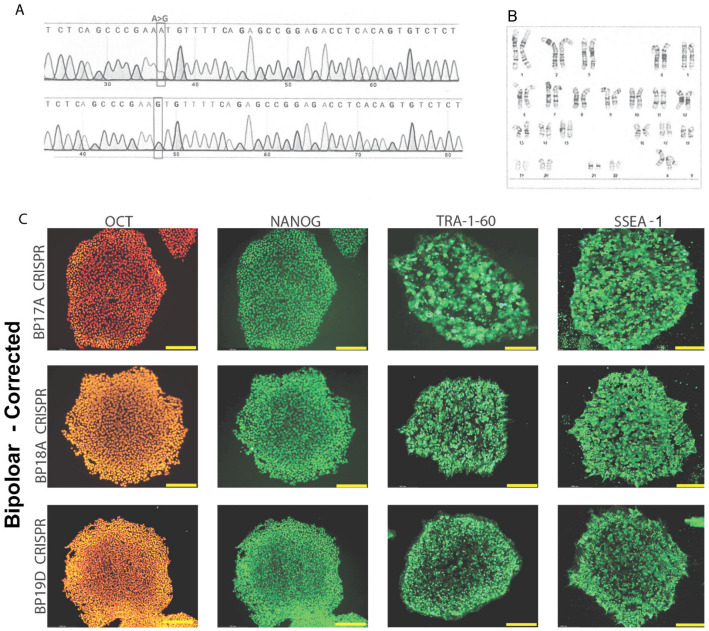
Generation and validation of CRISPR-corrected isogenic bipolar disorder cell lines. (**A**) Sanger sequencing representing CRISPR-mediated A to G conversion of rs*1006737* in a representative BP line. (**B**) Karyotype analysis of a representative BP-C line. (**C**) Representative immunocytochemistry images of the expression of pluripotency genes OCT4, NANOG, TRA-160 and SSEA-1 for each CRISPR-corrected bipolar disorder cell line, 20×. FITC-conjugated secondary antibodies.

**Figure 2 cells-13-01194-f002:**
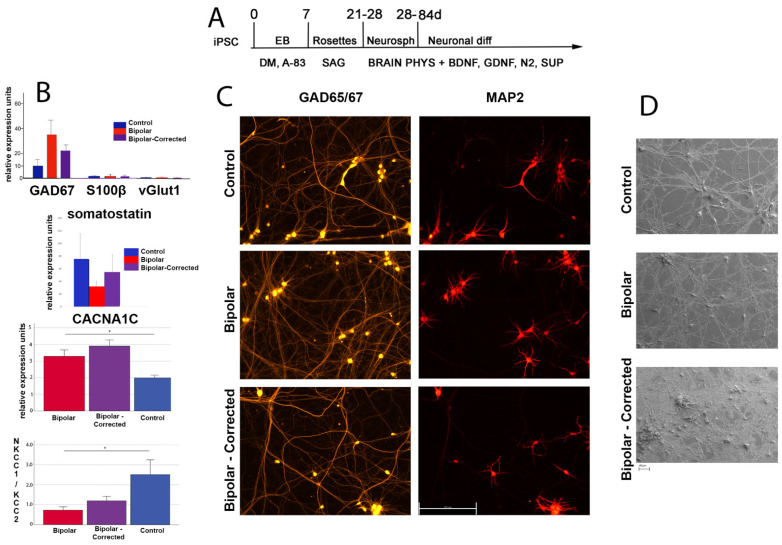
GABAergic neuron differentiation. (**A**) Protocol for the differentiation of GABAergic neurons from iPSC. (**B**) qRT-PCR analysis of the expression of the GABAergic marker GAD67, astrocyte-enriched transcript S100-β and marker of glutamatergic neurons _v_Glut1 in control, bipolar and bipolar-corrected neurons after 6 weeks of differentiation. Expressions of the GABAergic neuron subtype marker somatostatin, of CACNA1C and NKCC1/KCC2 mRNA in developing GABAergic neurons after 6 weeks of differentiation for control, bipolar and bipolar-corrected. * = *p* ≤ 0.05, Student’s *t*-test. Data from 6 independent differentiations were analyzed. (**C**) Immunocytochemical localization of the GABAergic marker, GAD65/67 and the neuronal protein, Map2, in representative control, bipolar and bipolar-corrected 6-week GABAergic neurons, 20X. (**D**) Representative scanning electron microscope (SEM) images of control, bipolar, and BP-C following 6 weeks of differentiation into GABAergic neurons, illustrating well-developed fields of processes and scattered cell bodies in BP-corrected cultures.

**Figure 3 cells-13-01194-f003:**
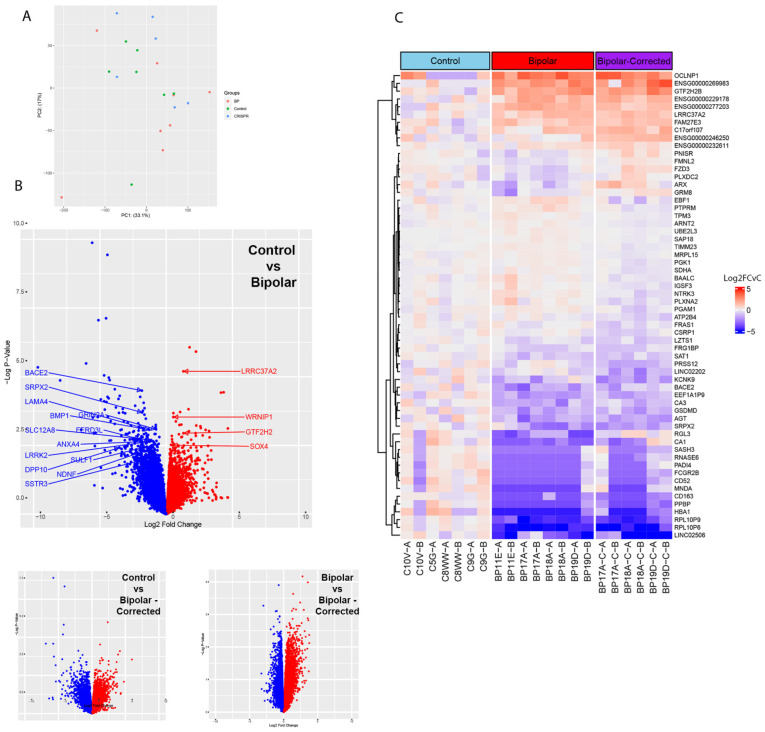
Bulk RNA sequencing analysis of GABAergic neurons. (**A**) Principal component analysis (PCA) plot of bipolar (BP), control and CRISPR-treated bipolar GABAergic neuron samples at 6 weeks post-differentiation. Two independent differentiations per line were employed. Titles of the axes indicate the proportion of variation each principal component accounts for. (**B**) Volcano plots compare differentially expressed genes between control vs. bipolar, highlighting select genes from the top 250 differentially expressed genes. Genes more highly expressed in BP are shown in red, and those more highly expressed in controls are shown in blue. Corresponding plots for control vs. bipolar-corrected and bipolar vs. bipolar-corrected are also shown. (**C**) Heatmap of the top 25 differentially expressed genes from pairwise comparisons between control, bipolar and bipolar-corrected groups. The values represent log2-fold changes relative to the average control sample.

**Figure 4 cells-13-01194-f004:**
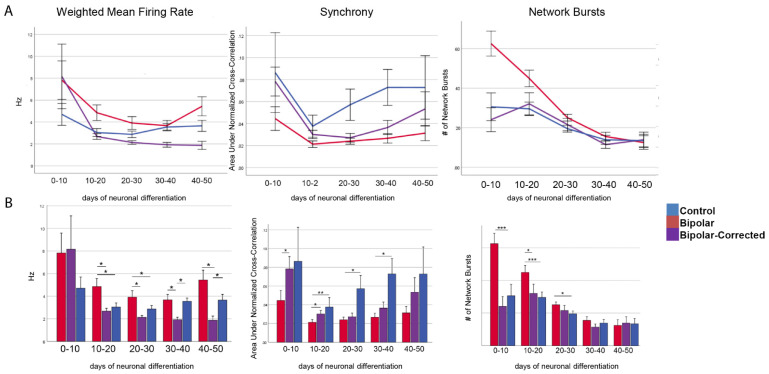
Multi-electrode array analysis of GABAergic neurons over 6 weeks of differentiation. Weighted mean firing rate, network bursting and synchrony were assessed from d0–d50 in vitro. Initially, bipolar neurons exhibited greater firing rates and network bursting compared to control neurons, while firing synchrony was higher in control neurons throughout the culture period. Activity of control, bipolar and BP-corrected GABAergic neurons over time are represented as line graphs (**A**) or bar graphs (**B**). * = *p* < 0.05, ** = *p* < 0.01, *** = *p* < 0.005, Student’s *t*-tests. Data from 7 independent replicates were analyzed.

**Figure 5 cells-13-01194-f005:**
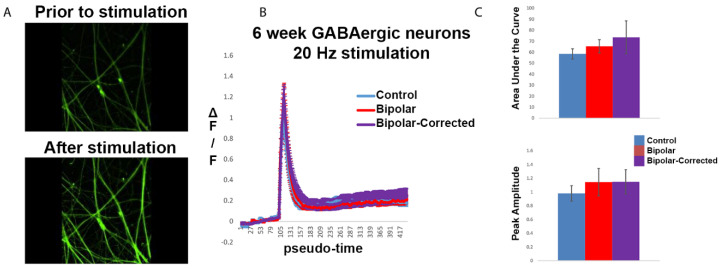
Live cell calcium imagining of GABAergic neurons. Six-week GABAergic neurons were subjected to 20 Hz stimulation and Ca^++^ handling was assessed using Fluo-5 imaging. (**A**) Representative images of control neurons pre- and post-stimulation are represented as line (**B**) and bar graphs (**C**), 20X. Control, bipolar and bipolar-corrected lines exhibited similar Ca^++^ handling patterns. Data from 11 independent differentiations are included.

**Figure 6 cells-13-01194-f006:**
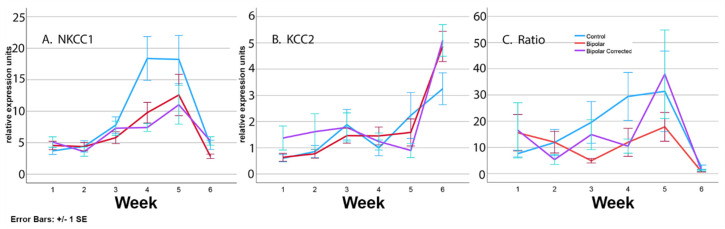
Expression of NKCC1 and KCC2 in bipolar disorder GABAergic neurons derived from BP iPSC. qRT-PCR analysis of the relative expression units of NKCC1 (**A**), KCC2 (**B**) and ratio of NKCC1 over KCC2 (**C**) in control, bipolar and bipolar-corrected GABAergic neurons over development. Data from 11 independent differentiations were analyzed and significance assessed using Student’s *t*-test and ANOVA.

**Table 1 cells-13-01194-t001:** Donor characteristics.

Lab Number	M/F	Age of Donor	rs1006737
C5	M	Newborn	GG
C8	M	59	GG
C9	F	49	GG
C10	F	32	GG
BP11	M	39	AA
BP17	F	26	AA
BP18	F	33	AA
BP19	M	59	AA

## Data Availability

Raw RNA-SEQ data and counts matrix are available at: GEO GSE 272060.

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
