# Peer review of "Human-Induced Pluripotent Stem Cell (iPSC)-Derived GABAergic Neuron Differentiation in Bipolar Disorder"

_cells, 2024, doi:10.3390/cells13141194_

Round 1

Reviewer 1 Report

Comments and Suggestions for Authors

This paper exploits GABAergic interneurons differentiated from induced pluripotent stem cells (iPSCs) of Bipolar (BP) and control (C) patients to investigate whether these in vitro models, alongside CRISPR-Cas9 corrected BP cells where the common SNP (rs1006737 A) in BP and schizophrenia (present in all BP patients considered in the study) is converted to the wild type nucleotide (G), display differences in GABAergic markers, morphology, immunohistochemistry (IHC), electrophysiological responses, calcium transients, and NKCC1 and KCC2 expression along with maturation and relative ratio.

The manuscript presents a well-conducted study that offers valuable insights into the potential roles of GABAergic interneurons and chloride transporters in Bipolar Disorder. The use of advanced techniques and the robust methodology are commendable. Moreover, the validation of patient iPSCs-derived excitatory and inhibitory neurons as an in vitro model of psychopathology has significant relevance in molecular psychiatry.

Some sections require minor revisions to improve clarity and coherence. Specifically, the introduction should better explain the advantages of using iPSCs, and the discussion should address the study limitations and potential implications more thoroughly. Additionally, the general absence of strong findings should be acknowledged, though this does not diminish the study interest.

Specific Comments:

Introduction Section:

I suggest adding more references in the introduction to better support the background information provided and to enhance the context of the study.

Cell Morphology:

The cells presented in Figure 2, specifically in the IHC and SEM images, exhibit well-developed and very long neuronal processes that appear more characteristic of excitatory neurons. Typically, GABAergic neurons have a more compact and rounded morphology with shorter dendrites, whereas glutamatergic neurons display a more elongated and extensively branched dendritic tree. Despite the expression of GABAergic markers such as GAD65/67 and the neuronal marker MAP2, the observed morphology raises questions about the predominant neuronal type in the cultures, especially given the absence of staining for parvalbumin, somatostatin, calretinin, neuropeptide Y, and cholecystokinin. To address this concern, I suggest the authors provide additional evidence supporting the GABAergic identity of these neurons.

Typos and Formatting:

Line 356: Correct the typo "bot"

Spacing:

Correct double and triple spacing after full stops throughout the manuscript

Comments on the Quality of English Language

Too many "there is" in the results section.

Reviewer 2 Report

Comments and Suggestions for Authors

The submission entitled „ Human induced pluripotent stem cell (iPSC)-derived GABAergic neuron differentiation and function in bipolar disorder" describes differences in maturation, network activity and gene expression between GABAergic (somatostatin+) neurons derived from human skin fibroblasts iPSCs of healthy subjects and individuals with bipolar disorder (BP) characterized by the single nucleotide polymorphysm (SNP), 75 rs1006737, in the L-type calcium channel gene CACNA1C. With calcium imaging this study does not detect a difference between normal and „BP" neurons in calcium currents induced by electrical stimulation.  Fig.2 shows increased expression of CACNA1C in „BP" neurons. Authors report increased firing, network bursting and decreased synchrony of network activity in cultures of „BP" neurons and different from the control timing of expression of NKCC1/KCC2.   The BP phenotype of GABAergic neurons could be corrected in this study by gene therapy (exchange A for G with CRISPR method).
The findings of this study are important and timely. I have only few minor comments and suggestions.

1)    Lettering of plots in Fig.2(B,D);  Fig3. (A,B); Fig.5(B,C), Fig.6 are by far too small. Graphics and their lettering need to be improved in quality.
2)    Sanger sequencing chromatogram shown in Fig 1A should be provided in color, not black/white.
3)    Abbreviations should be uniform throughout the text and Figure legends (e.g. BP-C for BP-corrected). Why „BP“, not „BD“ (for Bipolar Disorder), is used?
4)    Line 439: „neurons differentiated from BP iPSC are characterized by reduced synaptic densities (not shown)". These data should be shown.
5)    It is stated once in the text that only somatostatin-positive neurons were generated from iPSC by the method, not parvalbumin-positive. This should be stated also in the abstract and in the discussion.
6)    Line 266: „not parvalbumin (not shown)". These data should be shown.
7)    Line 273: two times „6-week"
8)    In primary dissociated cultures of neostriatum comprised of  GABAergic neurons, network activity is driven by glutamate/glutamine (it is blocked by D-AP5 and CNQX, PMID 28065671). Please show, how activity of BP, C or BP-C network is affected by these glutamate receptor antagonists. If activity is abolished, previous studies showing impairement in glutamatergic transmission and glutamate receptors in BP should be cited (PMID35544990).

Reviewer 3 Report

Comments and Suggestions for Authors

Schill, D. et al. Human induced pluripotent stem cell (iPSC)-derived GABAergic neuron differentiation and function in bipolar disorder. Cells 2024, 13, x. https://doi.org/10.3390/xxxxx

The manuscript is concerned with bipolar disorder (BP) patient-derived iPSCs which were differentiated into SST-expressing GABAergic interneurons. The authors found that BP neurons showed increased firing and network bursting with decreased synchrony. Their findings suggest that perturbations in GABAergic signaling during differentiation could contribute to cortical dysfunction in BP.

The topic is highly relevant to the field since the importance of inhibition for functional cell culture models is increasingly coming into focus and more reliable and improved protocols for the differentiation of GABAergic interneurons and specifically different interneuron subtypes are urgently needed. 

The text is written in a clear and comprehensible English.

The contents are decent and interesting but not ground-breaking.

Major general concerns: 

- The simplification and partially incorrect introduction to interneuron types significantly limits the transferability and applicability of the gained cell culture-based mechanistic findings to behavioral or clinical models. This generalization needs to be removed from the text and the specificity to SST-expressing interneurons needs to be emphasized and discussed.

- I would consider changing the manuscript title to "Human iPSC-Derived Somatostatin-Expressing GABAergic Neuron Differentiation and Function in Bipolar Disorder" to clarify that the differentiation protocol seems to have the capability to specifically create SST-expressing interneurons while not being generalizable to other subtypes. Without this addition, the impact of the applied methods and the resulting findings would be misleading.

- While the isolated, interneuron monoculture approach comes with a number of advantages, I wonder whether the observed effects would look very different in a more physiologically relevant co-culture with added excitatory components compared to the utilized purely GABAergic monoculture? Were any experiments in that direction conducted?

-In the pdf version I received, the figure resolution is not consistently satisfactory for publication (e.g. Figure 2B/D, 3A). This needs to be adjusted.

Line-specific comments:

Line 20: Replace "these cells" with a less vague term that extends beyond interneurons.

Line 51-55: This paragraph is incorrectly simplified and incomplete in the current form: "GABAergic interneurons form in the Medial Ganglionic Eminence (MGE) of the ventral telencephalon, [...], producing the inside-outside organization of the neocortex [5]."

--> It has long been known that GABAergic interneurons also form in other GE regions:  “Subregions of the GEs — the caudal GE (CGE), medial GE (MGE) and lateral GE (LGE) — generate distinct interneuron subtypes". (Eichmüller OL, Knoblich JA. Human cerebral organoids - a new tool for clinical neurology research. Nat Rev Neurol. 2022 Nov;18(11):661-680. doi: 10.1038/s41582-022-00723-9. Epub 2022 Oct 17. PMID: 36253568; PMCID: PMC9576133.) Moreover, from the same source: 

"Similarly, the distribution of interneuron subtypes differs. MGE-derived interneurons are found in deep layers across species and populate the mouse cortex uniformly, but CGE-derived interneurons are enriched in the upper layers in primates and humans (Fig. 2d), more so in areas associated with higher cognition, such as the prefrontal cortex, where they account for up to 50% of all interneurons"

- The distinction which type of interneuron was created is highly relevant to how they influence bipolar disorder since "Each group [of interneurons] includes several [sub]types [...] that differ in morphological and electrophysiological properties and likely have different functions in the cortical circuit." (Rudy et al. (2011). Three groups of interneurons account for nearly 100% of neocortical Gabaergic neurons. Dev. Neurobiol. 71, 45–61. doi: 10.1002/dneu.20853)

-Relatively new evidence further suggests the existence of locally born cortical interneurons formed from the same cortical progenitor cells which primarily generate excitatory neurons. (Delgado et al. (2022). Individual human cortical progenitors can produce excitatory and inhibitory neurons. Nature 601, 397–403. doi: 10.1038/s41586-021-04230-7). 

--> However, in the manuscript only SST-expressing interneurons were detected (quote from line 265: "GABAergic neurons robustly expressed SST (Figure 2B), but not other GABAergic neuron subtype markers such as parvalbumin"). For this subgroup, the MGE is the primary but still not the sole contributor! 

"Somatostatin (SST)-containing interneurons are primarily generated in the MGE, but could also derive from the ventral caudal ganglionic eminence (vCGE) and the dorsal CGE (dCGE)." (Wonders, C., Anderson, S. The origin and specification of cortical interneurons. Nat Rev Neurosci 7, 687–696 (2006). https://doi.org/10.1038/nrn1954)

Regardless, in rodents, SST INs only account for 30-40% of all interneurons, so all mentioned areas are relevant and the generalized statement in line 51 has to be rephrased.

[Another supporting source: Ma et al. (2013). Subcortical origins of human and monkey neocortical interneurons. Nat Neurosci 16, 1588–1597 (2013). https://doi.org/10.1038/nn.3536]

Line 59-60: This is phrased as if the switch would only occur in interneurons. Please clarify. The GABAergic polarity switch is a general developmental process affecting various types of neurons, including both interneurons and excitatory neurons. (Cf. Ben-Ari Y et al. GABA: a pioneer transmitter that excites immature neurons and generates primitive oscillations. Physiol Rev. 2007 Oct;87(4):1215-84. doi: 10.1152/physrev.00017.2006. PMID: 17928584. or Heesen SH and Köhr G (2024) GABAergic interneuron diversity and organization are crucial for the generation of human-specific functional neural networks in cerebral organoids. Front. Cell. Neurosci. 18:1389335. doi: 10.3389/fncel.2024.1389335)

Line 109: The GABAergic differentiation method seems to be reproducible from the supplied information.  The verification of neuronal identity after GABAergic differentiation appears plausible.

Line 216: MEA acquisition parameters are reasonable and replicable. The analysis later is simple but sufficient.

Line 247: The validation of CRISPR-corrected isogenic bipolar disorder cell lines seems plausible.

Line 284: Figure 2B.2-4: I would suggest finding a more visually elegant solution to the vertical writing of the markers, they are difficult to read. A simple 90° angle tilt of the text would probably work better. The subfigure alignment is partially off. The image resolution also needs to be increased.

Line 292: Figure 2C : As per convention, it would be advisable to add a scale bar to one of the images.

Round 2

Reviewer 3 Report

Comments and Suggestions for Authors

The authors modified the manuscript properly.